# Novel Treatment Strategies for Low-Risk Metastatic Castration-Sensitive Prostate Cancer

**DOI:** 10.3390/cancers16183198

**Published:** 2024-09-19

**Authors:** Hiroaki Iwamoto, Tomohiro Hori, Ryunosuke Nakagawa, Hiroshi Kano, Tomoyuki Makino, Renato Naito, Hiroshi Yaegashi, Shohei Kawaguchi, Takahiro Nohara, Kazuyoshi Shigehara, Kouji Izumi, Atsushi Mizokami

**Affiliations:** Department of Integrative Cancer Therapy and Urology, Kanazawa University Graduate School of Medical Science, Kanazawa 920-8641, Japan; tomohirohori0823@stu.kanazawa-u.ac.jp (T.H.); maik2810ryu@staff.kanazawa-u.ac.jp (R.N.); kanazawa_iimati@staff.kanazawa-u.ac.jp (H.K.); makinot@staff.kanazawa-u.ac.jp (T.M.); rnaito@staff.kanazawa-u.ac.jp (R.N.); hyassy@med.kanazawa-u.ac.jp (H.Y.); kawaguchi@staff.kanazawa-u.ac.jp (S.K.); nohara-kains@staff.kanazawa-u.ac.jp (T.N.); kshigehara0415@staff.kanazawa-u.ac.jp (K.S.); mizokami@staff.kanazawa-u.ac.jp (A.M.)

**Keywords:** metastatic castration-sensitive prostate cancer, androgen deprivation therapy, novel androgen receptor signaling inhibitor, combined androgen blockade, prostate-specific antigen, PSA reduction rate

## Abstract

**Simple Summary:**

Upfront novel androgen receptor signaling inhibitors (ARSIs) are the first-line treatment for metastatic castration-sensitive prostate cancer (mCSPC). However, there are a certain number of cases in which androgen deprivation therapy (ADT) is more effective in patients of Asian descent. If we can identify patients who show a marked response to ADT within 12 weeks after ADT, which is the inclusion criterion for upfront ARSI clinical trials, it would be valuable from an economic standpoint. A total of 218 patients who received ADT treatment at Kanazawa University Hospital between 2000 and 2020 were included in this study. Multivariate analysis revealed that a decrease in PSA levels of <95% at 12 weeks after ADT initiation was a predictor of short time to castration resistance (TTCR) in low-risk patients. We propose a new treatment strategy, in which patients with low-risk mCSPC are treated with ADT and switched to ARSIs, based on the rate of PSA reduction at 12 weeks.

**Abstract:**

Background: The treatment strategy for metastatic castration-sensitive prostate cancer (mCSPC) has changed significantly in recent years. Based on various guidelines, an upfront androgen receptor signaling inhibitor (ARSI) is the first choice, but in patients of Asian descent, including Japanese patients, there are a certain number of cases in which androgen deprivation therapy (ADT) and CAB are more effective. If patients can be identified who show a marked response to ADT within 12 weeks after the initiation of ADT, which is the inclusion criterion for ARSI clinical trials targeting mCSPC, it would be valuable from an economic standpoint. Methods: A total of 218 patients with pure prostate adenocarcinoma and treated with ADT at the Kanazawa University Hospital between January 2000 and December 2020 were included in this study. As a risk classification for mCSPC, in addition to the LATITUDE and CHAARTED criteria, we used the castration-sensitive prostate cancer classification proposed by Kanazawa University (Canazawa), developed by the Department of Urology of Kanazawa University. The Canazawa classification was based on three factors: Gleason pattern 5, bone scan index (BSI) ≥ 1.5, and lactate dehydrogenase (LDH) ≥ 300 IU/L. It defined patients with one factor or less as low-risk and patients with two or three factors as high-risk. The overall survival (OS) and time to castration resistance (TTCR) were estimated retrospectively using the Kaplan–Meier method, and factors associated with TTCR were identified using univariate and multivariate analyses. Results: The median follow-up period was 40.4 months, the median OS period was 85.2 months, and the median TTCR period was 16.4 months. The Canazawa risk classification provided the clearest distinction between the OS and TTCR in mCSPC patients. Multivariate analysis revealed a decrease in PSA levels of <95% at 12 weeks after ADT initiation and was a predictor of short TTCR in low-risk, low-volume patients across all risk classifications. Conclusion: The Canazawa classification differentiated the prognosis of mCSPC patients more clearly. A PSA reduction rate of <95% at 12 w after starting ADT in low-risk, low-volume patients of all risk classifications was significantly shorter than the TTCR. We propose a new treatment strategy, in which patients with low-risk mCSPC are treated with ADT and switched to ARSIs based on the rate of PSA reduction at 12 w.

## 1. Introduction

Prostate cancer (PC) is the most common cancer in men and the leading cause of cancer-related death in developed countries [1]. Approximately 10% of PC patients have distant metastases at diagnosis and have metastatic castration-sensitive prostate cancer (mCSPC) [2,3]. Following the establishment of androgen deprivation therapy (ADT) for PC in 1941, ADT targeting androgen receptor (AR) signaling has been the standard treatment for metastatic PC [4]. CAB (combined androgen blockade), consisting of a nonsteroidal antiandrogen plus either a luteinizing hormone-releasing hormone agonist (LHRH-A) or a bilateral orchiectomy, was developed to further improve prognosis over ADT monotherapy, but the prognostic benefit was unclear [5]. In recent years, the development of novel androgen receptor signaling inhibitors (ARSIs) has significantly changed the treatment regimen for mCSPC, with ARSIs being the first choice. Currently, ADT + ARSIs (either abiraterone, enzalutamide, or apalutamide) or ADT + docetaxel (DTX) + ARSIs (either abiraterone or darolutamide) are recommended for the treatment of mCSPC [6]. A certain number of cases have a long-term response to ADT or CAB (combined androgen blockade) in patients of Asian descent, including Japanese patients [7,8]. Therefore, administering an ARSI upfront to all low-risk mCSPC patients may be overtreatment. It is important from a medical and economic point of view to identify patients who have a significant response to ADT within 12 weeks after beginning ADT, which is the inclusion criterion for an ARSI clinical trial for mCSPC.

## 2. Materials and Methods

### 2.1. Patient Selection

Of the mCSPC patients who were treated at Kanazawa University Hospital between January 2000 and December 2020, 218 patients who were diagnosed with pure adenocarcinoma and treated with ADT were included in this study and evaluated retrospectively.

### 2.2. Collection of Clinical Data

All patient demographic, pathologic, and follow-up data, including the clinical stage, course of treatment, blood-draw data, castration-resistant prostate cancer (CRPC) progression date, and survival, were collected from medical records. The clinical stage was determined based on the 2017 TNM grading system, 8th edition [9]. Prostate-specific antigen (PSA) failure after ADT was defined as a 25% increase from the nadir, which was confirmed by a second PSA test at least four weeks later, with a PSA increase of at least 2.0 ng/mL. CRPC was diagnosed when the above criteria were met. All treatment regimens, blood tests, and imaging intervals following mCSPC diagnosis were determined by each attending physician. The follow-up for this study ended on 31 December 2023. The median follow-up period was 40.4 months. In addition to the LATITUDE and CHAARTED criteria [10,11], the castration-sensitive prostate cancer classification proposed by Kanazawa University (Canazawa) was used as the risk classification for mCSPC [12]. As shown in Figure 1, the Canazawa risk classification defined patients with zero factors or one factor as low-risk patients and patients with two or three factors as high-risk patients. The three factors included Gleason pattern 5, bone scan index (BSI) ≥ 1.5, and lactate dehydrogenase (LDH) ≥ 300 IU/L [12]. In the original study, a factor of zero was defined as low-risk, a factor of one as intermediate-risk, and factors of two or three as high-risk [12].

### 2.3. Statistical Analyses

The overall survival (OS) and time to castration resistance (TTCR) were estimated using the Kaplan–Meier method, and survival distributions were compared using the log-rank test. The hazard ratio (HR) and 95% confidence intervals (CI) were calculated. Multivariate analysis was conducted using the Cox proportional hazards model. For statistical analysis, commercially available SPSS software, version 25.0 (SPSS Inc., Chicago, IL, USA) and Prism v.9 (GraphPad, San Diego, CA, USA) were used. For all analyses, a *p*-value < 0.05 was considered statistically significant.

### 2.4. Ethical Considerations

This study was approved by the Institutional Review Board of Kanazawa University Hospital (2016-328). Informed consent was obtained in the form of an opt-out form posted at our facility which was approved by the Medical Ethics Committee of Kanazawa University. All methods were performed in accordance with relevant guidelines and regulations.

## 3. Results

### 3.1. Patient Characteristics

Table 1 shows the characteristics of the 218 mCSPC patients included in this study. The median age at diagnosis of mCSPC was 71.5 years (range 43–95), the median PSA value was 239 (2.1–16,701) ng/mL, the median LDH value was 189 (116–934) IU/L, and the median alkaline phosphatase (ALP) value was 353 (134–13,293) IU/L. Of the 218 patients, 215 underwent combined androgen blockade (CAB), 1 underwent surgical castration, and the treatments of 2 were unknown. In the study cohort, 183 (83.9%) patients had a GS of eight or higher, 25 (11.5%) had M1c, and 160 (73.4%) ultimately progressed to CRPC. All-cause mortality during the follow-up period was observed in 98 patients (45.0%), and PC-specific mortality was observed in 81 patients (37.2%). There were 66 LATITUDE low-risk patients (31.6%), 70 CHAARTED low-volume patients (33.5%), and 91 Canazawa low-risk patients (68.9%). Patients with missing data were excluded for the risk classification.

### 3.2. Kaplan–Meier Curves of Overall Survival

Kaplan–Meier survival curves for the OS in this cohort are shown in Figure 2A–D. As shown in Figure 2A, the median OS for all patients was 85.2 months. The OS by LATITUDE criteria is shown in Figure 2B. The median OS for the LATITUDE low-risk group was 135.0 months, whereas it was 56.1 months for the high-risk group, which was significantly shorter (HR: 2.54, *p* < 0.0001). The OS by CHAARTED criteria is shown in Figure 2C. The median OS period for the CHAARTED low-volume group was 135.0 months and 55.1 months in the high-volume group, which was significantly shorter (HR: 2.78, *p* < 0.0001). The OS based on the Canazawa risk classification is shown in Figure 2D. The Canazawa low-risk group had a median OS period of 135.0 months, whereas the high-risk group had a median OS period of 38.0 months, which was significantly shorter (HR: 3.67, *p* < 0.0001). The Canazawa risk classification most clearly differentiated the OS.

### 3.3. Kaplan–Meier Curves of Time to Castration Resistance

Kaplan–Meier survival curves for TTCR are shown in Figure 3A–D. As shown in Figure 3A, the median TTCR period for all patients was 16.4 months. The TTCR by LATITUDE criteria is shown in Figure 3B. The median TTCR for the LATITUDE low-risk group was 28.9 months and it was 12.9 months for the high-risk group, which was significantly shorter (HR: 1.85, *p* = 0.0004). The TTCR by CHAARTED criteria is shown in Figure 3C. The median TTCR in the CHAARTED low-volume group was 24.7 months, and it was 12.3 months in the high-volume group, which was significantly shorter (HR: 1.68, *p* = 0.0003). The TTCR divided by the Canazawa risk classification is shown in Figure 3D. The median TTCR for the Canazawa low-risk group was 135.0 months and 38.0 months for the high-risk group, which was significantly shorter (HR: 2.69, *p* < 0.0001). Thus, the Canazawa risk classification differentiated the TTCR the most.

### 3.4. Identification of Prognostic Factors in Time to Castration Resistance in Low-Risk or Low-Volume mCSPC

In the LATITUDE low-risk, CHAARTED low-volume, and Canazawa low-risk groups, multivariate analysis was performed using a Cox proportional hazards model to determine whether changes in serum markers at 4 and 12 weeks after the start of ADT can predict TTCR. Three serum markers, PSA, LDH, and ALP, were used. For all risk classifications, no significant factors were found that predicted TTCR after four weeks of ADT (Appendix A). Table 2 shows the results of a multivariate analysis to determine whether TTCR could be predicted 12 weeks after the start of ADT in the LATITUDE low-risk group. Multivariate analysis indicated that an independent significant predictor of worse TTCR was a PSA reduction rate of less than 95% after 12 weeks of ADT (HR: 4.70, 95% CI 1.39–15.88; *p* = 0.018). A multivariate analysis was conducted to determine whether TTCR could be predicted after 12 weeks of ADT in the CHAARTED low-volume group (Table 3). The results indicated that an independent and significant predictor of worse TTCR was a reduction in PSA levels of <95% after 12 weeks of ADT (HR: 3.57, 95% CI 1.17–10.95; *p* = 0.026). Next, a multivariate analysis was conducted to determine whether TTCR could be predicted after 12 weeks of ADT in the Canazawa low-risk group (Table 4). Multivariate analysis showed that a significant independent predictor of worse TTCR was a PSA reduction rate of <95% after 12 weeks of ADT (HR: 2.66, 95% CI 1.29–5.50; *p* = 0.008).

### 3.5. Kaplan–Meier Curve of Time to Castration Resistance in Low-Risk or Low-Volume mCSPC by PSA Reduction Rate after 12 Weeks of ADT

The Kaplan–Meier curve for TTCR in low-risk or low-volume mCSPC groups based on the rate of PSA decline after 12 weeks of ADT is shown in Figure 4A–C. The TTCR in the LATITUDE low-risk group based on the rate of PSA decline after 12 weeks of ADT is shown in Figure 4A. The median TTCR based on the group with a PSA reduction rate of >95% was 34.3 months versus 16.1 months in the group with a PSA reduction rate of <95%, which was significantly shorter (HR: 2.18, *p* = 0.041). The TTCR of the CHAARTED low-volume group based on a PSA reduction rate after 12 weeks of ADT is shown in Figure 4B. The median TTCR for the group with a PSA reduction rate of >95% was 28.9 months versus 16.1 months in the group with a PSA reduction rate of <95%, which was significantly shorter (HR: 2.14, *p* = 0.038). The TTCR for the Canazawa low-risk group based on a PSA reduction rate after 12 weeks of ADT is shown in Figure 4C. The median TTCR in the group with a PSA reduction rate of >95% was 28.9 months versus 13.1 months in the group with a PSA reduction rate of <95%, which was significantly shorter (HR: 2.27, *p* = 0.0073).

## 4. Discussion

In the present study, the Canazawa risk classification, rather than the LATITUDE or CHAARTED criteria, clearly differentiated the OS in mCSPC patients who underwent ADT. Increasing the number of patients in the cohort used to establish the Canazawa risk classification and extending the observation period did not change the results of the Canazawa risk classification. The Canazawa risk classification consists of three factors: Gleason pattern 5, BSI ≥ 1.5, and LDH ≥ 300 IU/L [12]. The contribution of bone involvement in the risk stratification of mCSPC patients has been widely reported. For the LATITUDE criteria, the presence of three or more bone lesions was considered a high-risk factor, and in the CHAARTED criteria, the presence of four bone lesions, including one over the vertebral body and pelvis, was considered a high-volume factor [10,11]. An EOD 4 and EOD 3 or higher are also considered to be prognostic factors for the OS in the mCSPC risk classification in Japanese patients [13,14]. The EOD score is based on the number of bone lesions and does not consider the size of the bone lesions. The BSI is an objective marker of the number of bone metastases throughout the body, rather than a subjective number [15,16]. The Canazawa risk classification is the first risk classification that uses BSI. The Canazawa risk classification may be an excellent tool to objectively evaluate the prognosis of mCSPC.

The results of large international clinical trials in mCSPC have shown that upfront ARSI administration prolongs the OS [17,18,19]; however, there are scattered reports in Japanese real-world data showing that upfront ARSI administration prolongs PFS, but does not significantly differ in OS [20,21,22]. In a retrospective report of 178 patients with LATTITUDE high-risk mCSPC, abiraterone plus ADT significantly prolonged PFS compared with bicalutamide plus ADT, but the OS was not significantly different [20]. In another retrospective report of 312 patients with LATTITUDE high-risk mCSPC, the abiraterone + ADT group significantly prolonged the time to CRPC development compared with the bicalutamide + ADT group, but the OS was not significantly different [21]. In a retrospective study of 581 mCSPC patients, including 442 LATTITUDE high-risk and 139 low-risk cases, the ARSI + ADT group significantly prolonged PSA-PFS compared with the bicalutamide + ADT group [22]. However, only the high-risk group showed a significant difference in OS, and no significant difference in OS was observed in general. The lack of a significant difference between the ARSI and ADT groups in the Japanese real-world data may be the result of racial factors. In a retrospective study of 165 PC patients undergoing ADT in Honolulu, Japanese American men who received ADT had a better prognosis compared with white men in terms of both overall survival and cause-specific survival [8]. Multivariate analysis also revealed that race was an important prognostic factor. A retrospective study compared data from the USA Cancer of the Prostate Strategic Urologic Research Endeavor (CaPSURE) registry database and the Japanese Cancer of the Prostate (J-CaP) registry database for a total of 15,513 PC patients undergoing ADT [7]. Multivariate regression analysis showed that the sub-hazard ratio for prostate cancer-specific mortality (PCSM) in J-CaP and CaPSURE was 0.52, indicating that the adjusted PCSM of men who received ADT in Japan was less than half that of men in the United States. Although further large-scale prospective studies are pending, it is anticipated that patients of Asian descent, including Japanese patients, will benefit more from ADT compared with patients of Caucasian descent. In light of the above, administering ARSIs upfront to all low-risk mCSPC patients may be overtreatment. The cost of administering upfront ARSIs has been reported to be 10.6 to 30.8 times higher than conventional therapy, which is undesirable from a health economic perspective [23]. Cost is not the only disadvantage of administering upfront ARSIs to all patients. We have shown that the use of ARSIs may induce the appearance of visceral metastases and neuroendocrine differentiation (NED) [2]. A subanalysis of the TITAN study also reported a higher percentage of image progression without PSA elevation in the apalutamide group compared to the ADT group [24]. These patients had a higher rate of visceral metastases, a characteristic of neuroendocrine prostate cancer (NEPC). NEPC has a poor prognosis, and no effective treatment has been established [25]. In summary, ARSI administration may induce NED, and blindly administering upfront ARSIs to all patients may be overtreatment.

Many studies have reported that PSA nadir and time to PSA nadir (TTN) are factors that predict the prognosis of prostate cancer in patients after ADT [26,27,28]; however, PSA nadir and TTN require long-term observation and are not useful for early prognosis prediction. The PSA reduction rate at 12 weeks after the start of ADT identified in the present study can predict the prognosis of low-risk mCSPC early. Identifying patients who do not require upfront ARSI administration is useful; however, the benefits of upfront ARSI administration should not be lost as a result. Originally, upfront ARSIs were approved for ADT administration within 12 weeks. To identify patients who do not require upfront ARSIs, it is necessary to identify patients will respond to ADT in the long term within a 12-week period. If we can distinguish patients who will respond to ADT in the long-term from those who will not within 12 weeks, it will be possible to switch to ARSIs. In the present study, the group with a PSA reduction rate of <95% at 12 weeks after starting ADT had a significantly shorter TTCR in all risk categories. These cases can be switched to ARSIs at 12 weeks and treated as upfront ARSIs. The group with a PSA reduction rate of ≥95% at 12 weeks after starting ADT had a median TTCR period of 24.7–28.9 months, and long-term responses could be expected. Thus, ADT could be continued and the treatment switched to ARSIs if CRPC developed. We propose the above treatment strategy for low-risk mCSPC. The present study had several limitations. All patients were of Japanese descent and were analyzed retrospectively. In addition, PC treatment was left to the discretion of the attending physician, which may have resulted in bias. Larger prospective studies are needed to confirm our findings.

## 5. Conclusions

In this study, the Canazawa classification most clearly differentiated the prognosis of mCSPC patients. A PSA reduction rate of <95% at 12 W after starting ADT in low-risk, low-volume patients of all risk classifications had a significantly shorter TTCR. We propose a new treatment strategy in which patients with low-risk mCSPC are started on ADT and switched to ARSIs based on the rate of PSA reduction at 12 W.

## Figures and Tables

**Figure 1 cancers-16-03198-f001:**
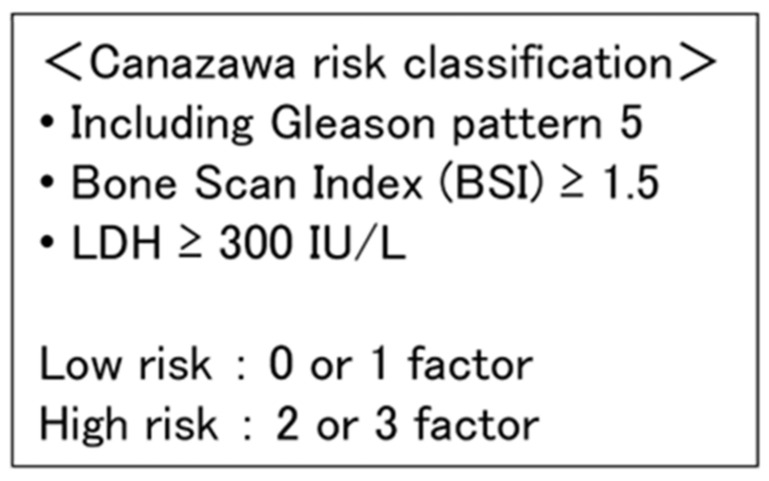
Definition of the castration-sensitive prostate cancer classification proposed by Kanazawa University (Canazawa).

**Figure 2 cancers-16-03198-f002:**
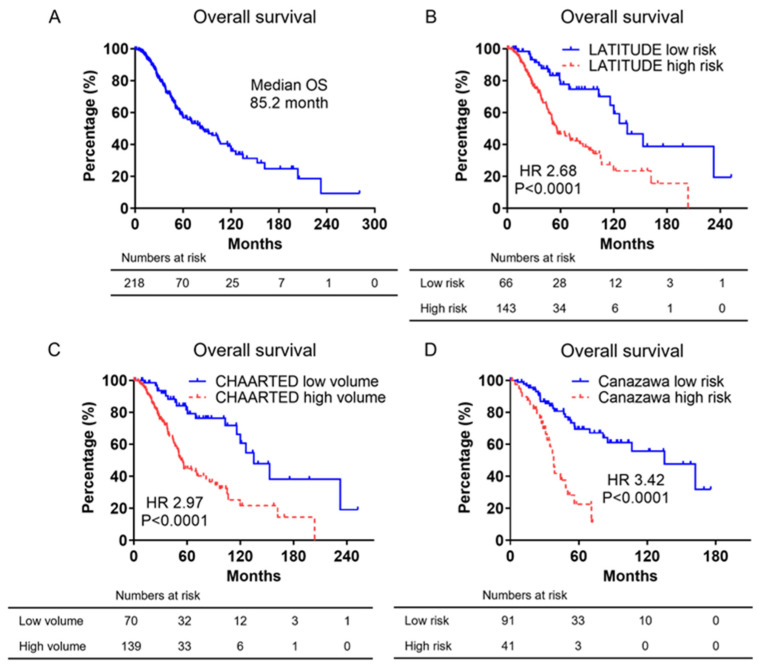
Kaplan–Meier curves showing OS in mCSPC patients. (**A**) Kaplan–Meier curves of OS for all patients. (**B**) Kaplan–Meier curves of OS divided by LATITUDE criteria. (**C**) Kaplan–Meier curves of OS divided by CHAARTED criteria. (**D**) Kaplan–Meier curves of OS divided by Canazawa risk classification.

**Figure 3 cancers-16-03198-f003:**
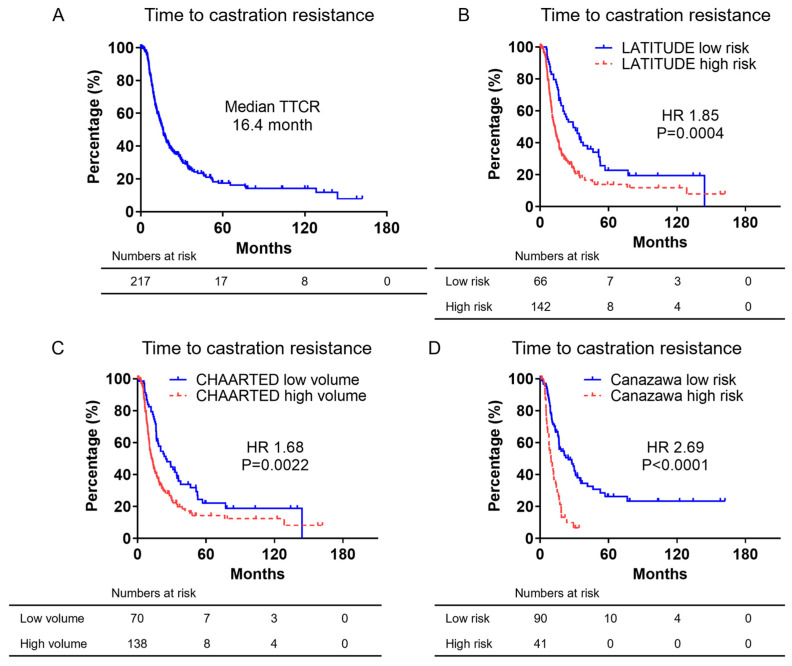
Kaplan–Meier curves showing TTCR in mCSPC patients. (**A**) Kaplan–Meier curves of TTCR for all patients. (**B**) Kaplan–Meier curves of TTCR divided by LATITUDE criteria. (**C**) Kaplan–Meier curves of TTCR divided by CHAARTED criteria. (**D**) Kaplan–Meier curves of TTCR divided by Canazawa risk classification.

**Figure 4 cancers-16-03198-f004:**
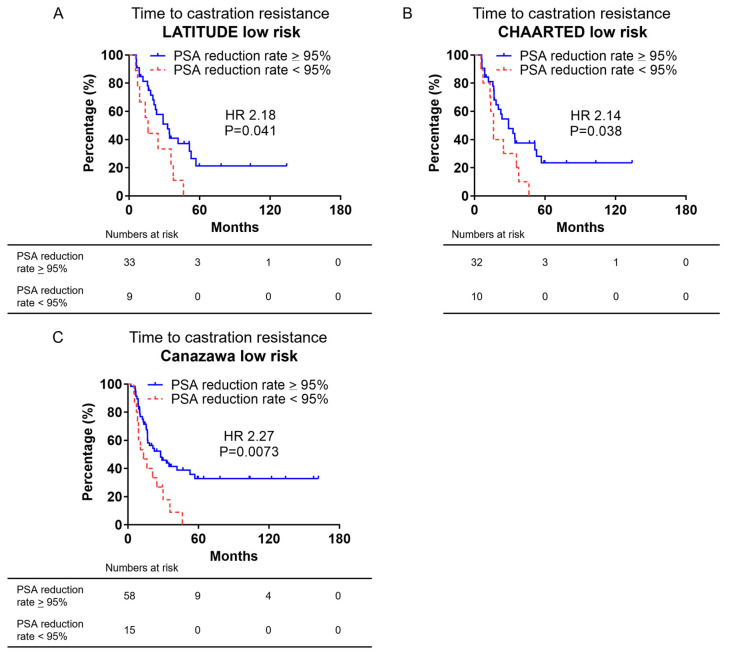
Kaplan–Meier curves showing TTCR in low-risk or low-volume mCSPC groups. (**A**) Kaplan–Meier curves of TTCR divided by PSA reduction rate after 12 weeks of ADT in LATITUDE low-risk patients. (**B**) Kaplan–Meier curves of TTCR divided by PSA reduction rate after 12 weeks of ADT in CHAARTED low-volume patients. (**C**) Kaplan–Meier curves of TTCR divided by PSA reduction rate after 12 weeks of ADT in Canazawa low-risk patients.

**Table 1 cancers-16-03198-t001:** Patient characteristics.

	LATITUDE		CHAARTED		Canazawa		All
Characteristics	Low-Risk	High-Risk	Low-Volume	High- Volume	Low-Risk	High-Risk	
Patients, *n*	66	143	70	139	91	41	218
Median age, yr	71.5	71.0	70.0	71.0	71	73	71.5
Median PSA, ng/mL	103.7	331.8	117.0	274.5	200.5	851.8	239
Median LDH, IU/L	178.5	194.5	180	191	186	212	189
Median ALP, IU/L	232	408	262.5	403.5	271	623	353
Histology, *n* (%)							
GS ≤ 7	19 (28.8)	10 (7.0)	15 (21.4)	14 (10.1)	16 (17.6)	0	32 (14.7)
GS ≥ 8	45 (68.2)	132 (92.3)	53 (75.7)	124 (89.2)	75 (82.4)	41 (100)	183 (83.9)
Unknown	2 (3.0)	1 (0.7)	2 (2.9)	1 (0.7)	0	0	3 (1.4)
T stage, *n* (%)							
T1–2	8 (12.1)	10 (7.0)	6 (8.6)	12 (8.6)	10 (11.0)	4 (9.8)	21 (9.6)
T3	20 (30.3)	68 (47.6)	23 (32.9)	65 (46.8)	47 (51.6)	16 (39.0)	90 (41.3)
T4	31 (47.0)	51 (35.7)	32 (45.7)	50 (36.0)	29 (31.9)	18 (43.9)	85 (39.0)
Tx	7 (10.6)	14 (9.8)	9 (12.9)	12 (8.6)	5 (5.5)	3 (7.3)	22 (10.1)
N stage, *n* (%)							
N0	31 (47.0)	51 (35.7)	31 (44.3)	51 (36.7)	33 (36.3)	16 (39.0)	86 (39.4)
N1	35 (53.0)	88 (61.5)	38 (54.3)	85 (61.2)	57 (62.6)	25 (61.0)	128 (58.7)
Nx	0	4 (2.8)	1 (1.4)	3 (2.2)	1 (1.1)	0	4 (1.8)
M stage, *n* (%)							
M1a	6 (9.1)	1 (0.7)	6 (8.6)	1 (0.7)	2 (2.2)	0	7 (3.2)
M1b	59 (89.4)	118 (82.5)	64 (91.4)	113 (81.3)	78 (85.7)	37 (90.2)	186 (85.3)
M1c	1 (1.5)	24 (16.8)	0	25 (18.0)	11 (12.1)	4 (9.8)	25 (11.5)
Initial treatment, *n* (%)							
CAB	66 (100)	142 (99.3)	70 (100)	138 (99.3)	91 (100)	41 (100)	215 (98.6)
Surgical castration	0	1 (0.7)	0	1 (0.7)	0	0	1 (0.5)
Unknown	0	0	0	0	0	0	2 (0.9)
CRPC progression, *n* (%)	45 (68.2)	110 (76.9)	49 (70)	106 (76.3)	59 (64.8)	35 (85.4)	160 (73.4)
All-cause death, *n* (%)	21 (31.8)	74 (51.7)	21 (30.0)	74 (53.2)	30 (33.0)	26 (63.4)	98 (45.0)

PSA: prostate-specific antigen; LDH: lactate dehydrogenase; ALP: alkaline phosphatase; GS: Gleason score; CAB: combined androgen blockade; CRPC: castration-resistant prostate cancer.

**Table 2 cancers-16-03198-t002:** Univariate and multivariate analyses of TTCR in LATITUDE low-risk mCSPC (serum markers at 12 W after ADT).

			Univariate		Multivariate	
		*n*	HR (95% CI)	*p*	HR (95% CI)	*p*
PSA reduction rate, ng/mL	≥95%	33	2.44 (1.09–5.47)	0.03	4.70 (1.39–15.88)	0.013
	<95%	9				
LDH at 12 W after ADT, IU/L	<250 or reduction rate ≥ 30%	26	0.90 (0.26–3.15)	0.87	4.20 (0.996–17.69)	0.051
	≥250 and reduction rate < 30%	3				
ALP at 12 W after ADT, IU/L	<350 or reduction rate ≥ 30%	24	2.98 (0.83–10.72)	0.1	1.37 (0.33–5.65)	0.66
	≥350 and reduction rate < 30%	5				

TTCR: time to castration resistance; mCSPC: metastatic castration-sensitive prostate cancer; ADT: androgen deprivation therapy; PSA: prostate-specific antigen; LDH: lactate dehydrogenase; ALP: alkaline phosphatase; HR: hazard ratio; CI: confidence interval.

**Table 3 cancers-16-03198-t003:** Univariate and multivariate analyses of TTCR in CHAARTED low-volume mCSPC (serum markers at 12 w after ADT).

			Univariate		Multivariate	
		*n*	HR (95% CI)	*p*	HR (95% CI)	*p*
PSA reduction rate, ng/mL	≥95%	32	2.21 (1.02–4.78)	0.04	3.57 (1.17–10.95)	0.026
	<95%	10				
LDH at 12 W after ADT, IU/L	<250 or reduction rate ≥ 30%	25	2.73 (0.77–9.74)	0.12	3.46 (0.86–14.02)	0.082
	≥250 and reduction rate < 30%	3				
ALP at 12 W after ADT, IU/L	<350 or reduction rate ≥ 30%	22	1.01 (0.33–3.09)	0.98	1.18 (0.36–3.86)	0.79
	≥350 and reduction rate < 30%	6				

TTCR: time to castration resistance; mCSPC: metastatic castration-sensitive prostate cancer; ADT: androgen deprivation therapy; PSA: prostate-specific antigen; LDH: lactate dehydrogenase; ALP: alkaline phosphatase; HR: hazard ratio; CI: confidence interval.

**Table 4 cancers-16-03198-t004:** Univariate and multivariate analyses of TTCR in Canazawa low-risk mCSPC (serum markers at 12 W after ADT).

			Univariate		Multivariate	
		*n*	HR (95% CI)	*p*	HR (95% CI)	*p*
PSA reduction rate, ng/mL	≥95%	58	2.17 (1.16–4.05)	0.016	2.66 (1.29–5.50)	0.008
	<95%	15				
LDH at 12 W after ADT, IU/L	<250 or reduction rate ≥ 30%	61	1.91 (0.75–4.86)	0.18	2.28 (0.81–6.43)	0.12
	≥250 and reduction rate < 30%	6				
ALP at 12 W after ADT, IU/L	<350 or reduction rate ≥ 30%	60	1.13 (0.48–2.69)	0.77	1.11 (0.42–2.94)	0.84
	≥350 and reduction rate < 30%	8				

TTCR: time to castration resistance; mCSPC: metastatic castration-sensitive prostate cancer; ADT: androgen deprivation therapy; PSA: prostate-specific antigen; LDH: lactate dehydrogenase; ALP: alkaline phosphatase; HR: hazard ratio; CI: confidence interval.

## Data Availability

The data presented in this study are available in the article and Appendix A.

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
