# Peer review of "Novel Treatment Strategies for Low-Risk Metastatic Castration-Sensitive Prostate Cancer"

_cancers, 2024, doi:10.3390/cancers16183198_

Round 1
Reviewer 1 Report
Comments and Suggestions for Authors
July 22, 2024
Ms. Ref. No.: cancers-3118463
Journal: Cancers
Title: Novel Treatment Strategies for Low-Risk Metastatic Castration-Sensitive Prostate Cancer
Comments:
Thank you for taking the time to write this article on such a relevant topic. I found it to be informative and with the potential for further research in the future. However, I have some observations that I believe could be useful for improving the article, which I have mentioned in the following paragraphs.
1- According to this article, the androgen deprivation therapy (ADT) demonstrates greater efficacy in individuals of Asian. Can generalize this result for others? If yes, How?
2- The duration time of respond to ADT was about 12 weeks, why this 12 weeks?
3- The sample size of this study was A total of 218 patients, why this size and how was determine this size? Is it sufficient for responding to the first question?
4- Due to the methods and result section, is there any association between some important factors such as Obesity and weight &Family history.& Being taller and the result?
5- What was the inclusion and exclusion criteria for this research? (Age, Hormone levels,…,?)
6- According to the result section, PSA reduction rate is critical, is there another option for evaluating?
7- The timeframes of charts in Figure 2 are different. Up to 180, 240, 300 months, is it OK?
8- It seems to be better to edit section 3.1.1. Subsubsection , depends on authors guideline.
9- It is possible to enhance the introduction section's readability by including the following reference:
· https://doi.org/10.3390/cancers16142535
· https://doi.org/10.1007/s00210-023-02551-0
· https://doi.org/10.3390/cancers16132433
· https://doi.org/10.3390/cancers16142520
Author Response
Dear Samuel C. Mok and reviewers,
Thank you for reviewing my manuscript.
I have revised my manuscript according to the reviewers’ suggestions, and I believe that your comments will be helpful for improving quality of my manuscript.
The revised parts are shown in red.
Please re-review my manuscript.
Thanks again, and then I appreciate so much your kind efforts.
Yours sincerely,
Hiroaki Iwamoto, MD, PhD
Iwamoto-h@med.kanazawa-u.ac.jp
Department of Integrative Cancer Therapy and Urology
Kanazawa University Graduate School of Medical Science
Takaramachi, Kanazawa, Ishikawa, Japan
TEL: 076-265-2393, FAX 076-222-6726
<Reviewer: 1>
Thank you for taking the time to write this article on such a relevant topic. I found it to be informative and with the potential for further research in the future. However, I have some observations that I believe could be useful for improving the article, which I have mentioned in the following paragraphs.
- According to this article, the androgen deprivation therapy (ADT) demonstrates greater efficacy in individuals of Asian. Can generalize this result for others? If yes, How?
(Reply)
Thank you for kindly comments.
It is known that ADT shows higher efficacy in Asians.
We cannot generalize this effect to show higher efficacy in other races.
However, regardless of race, patients who meet a PSA reduction of 95 or greater after 12 weeks may be able to continue on ADT.
- The duration time of respond to ADT was about 12 weeks, why this 12 weeks?
(Reply)
Thank you for kindly comments.
In the ARSI clinical trial for mCSPC, ADT within 12 weeks prior to ARSI administration was allowed.
Therefore, we set 12 weeks as the criterion for determining the effectiveness of ADT.
This is described in the Introduction.
- The sample size of this study was A total of 218 patients, why this size and how was determine this size? Is it sufficient for responding to the first question?
(Reply)
Thank you for kindly comments.
Of the mCSPC patients treated at Kanazawa University Hospital from January 2000 to December 2020, 218 patients who were diagnosed with pure adenocarcinoma and started treatment with ADT were included and evaluated retrospectively.
Although a larger sample size is desirable, 218 patients were included in this cohort.
Further large prospective studies are desirable in the future.
- Due to the methods and result section, is there any association between some important factors such as Obesity and weight &Family history.& Being taller and the result?
(Reply)
Thank you for kindly comments.
In this study, we examined whether changes in blood collection data within 12 weeks after ADT could predict survival.
Weight and height do not change significantly over 12 weeks of ADT.
Therefore, we did not examine the factors you mentioned in this study.
In terms of sample size, it is difficult to subgroup by weight or height.
- What was the inclusion and exclusion criteria for this research? (Age, Hormone levels,…,?)
(Reply)
Thank you for kindly comments.
There are no exclusion criteria for this study.
Of the mCSPC patients treated at Kanazawa University Hospital from January 2000 to December 2020, 218 patients who were diagnosed with pure adenocarcinoma and started treatment with ADT were included and evaluated retrospectively.
- According to the result section, PSA reduction rate is critical, is there another option for evaluating?
(Reply)
Thank you for kindly comments.
In the present study, we examined PSA, ALP, and LDH, three serum markers that are well-known predictors of prostate cancer prognosis.
However, only PSA was able to predict prognosis based on changes after 12 weeks of ADT.
- The timeframes of charts in Figure 2 are different. Up to 180, 240, 300 months, is it OK?
(Reply)
Thank you for kindly comments.
This difference arises because patients with missing data were excluded from the risk classification.
I followed your advice and added the following sentence. "Patients with missing data were excluded for risk classification." ( P4, lines 104-105)
- It seems to be better to edit section 3.1.1. Subsubsection , depends on authors guideline.
(Reply)
Thank you for kindly comments.
I followed your advice and deleted the text.
9- It is possible to enhance the introduction section's readability by including the following reference:
- https://doi.org/10.3390/cancers16142535
- https://doi.org/10.1007/s00210-023-02551-0
- https://doi.org/10.3390/cancers16132433
- https://doi.org/10.3390/cancers16142520
(Reply)
Thank you for kindly comments.
The current study examined treatment strategies for mCSPC.
I find it difficult to add the literature against mCRPC to the intro.
If you have further advice on references to cite please let me know.
Reviewer 2 Report
Comments and Suggestions for Authors
Dear authors,
Friendly speaking, this study is totally unreadable based on the following reason:
1. Undisclosed unmet medical need. The authors did not disclose why you need to find new treatment for CRPC.
2. Confusing materials and methods. The Materials and Methods mixes the description of template and original description. Also, such description does not correlate to the rationale disclosed in the introduction.
3. Non-corresponded results. The results did not related to the rationale of this study.
Comments on the Quality of English LanguageThis study is hard to read.
Author Response
Dear Samuel C. Mok and reviewers,
Thank you for reviewing my manuscript.
I have revised my manuscript according to the reviewers’ suggestions, and I believe that your comments will be helpful for improving quality of my manuscript.
The revised parts are shown in red.
Please re-review my manuscript.
Thanks again, and then I appreciate so much your kind efforts.
Yours sincerely,
Hiroaki Iwamoto, MD, PhD
Iwamoto-h@med.kanazawa-u.ac.jp
Department of Integrative Cancer Therapy and Urology
Kanazawa University Graduate School of Medical Science
Takaramachi, Kanazawa, Ishikawa, Japan
TEL: 076-265-2393, FAX 076-222-6726
Reviewer's comments:
<Reviewer: 2>
Dear authors,
Friendly speaking, this study is totally unreadable based on the following reason:
- Undisclosed unmet medical need. The authors did not disclose why you need to find new treatment for CRPC.
(Reply)
Thank you for kindly comments.
As Asians, including the Japanese, have a high response rate to ADT, and performing upfront ARSI in all mCSPC patients may result in overtreatment.
It is also undesirable from a medical economic standpoint.
Furthermore, additional ARSI may induce NEDs.
Therefore, we believe it is important to identify patients with mCSPC who do not require upfront ARSI.
The above is described in detail in the Discussion. ( P11-12, lines 238-274)
- Confusing materials and methods. The Materials and Methods mixes the description of template and original description. Also, such description does not correlate to the rationale disclosed in the introduction.
(Reply)
Thank you for kindly comments.
I have followed your advice and removed the template description and modified it to make it more readable.
- Non-corresponded results. The results did not related to the rationale of this study.
Comments on the Quality of English Language
This study is hard to read.
(Reply)
Thank you for kindly comments.
I have followed your advice and have had the text edited by a native English speaker.
Reviewer 3 Report
Comments and Suggestions for Authors
This is an interesting paper that aimed to propose a cost-effective treatment strategy that also potentially avoids overtreatment. Based on outcome data from published studies and local patients, the authors applied the Canazawa criteria and stratified patients into low and high risk groups, and proposed in the low risk group a subset of patients can be safely kept on ADT before adding ARSI. Several issues need to be considered
Major comments
1. The authors stated patient can be treated with ADT and be switched to ARSI based on the rate of PSA reduction at 12 weeks. The only data was shown was patient had < 95% PSA reduction at 12 weeks has worse outcome. There is no data support by adding ARSI to these patients can improve their outcome comparing to only ADT alone.
2. The authors need to clarify how the threshold of 95% reduction of PSA was determined.
3. Figure 1 - total patients included in the study was 218. The sum of low risk and high risk patient numbers are less than 218 for the two trials and Canazawa criteria. Where are the missing patients?
4. The introduction part needs to be expanded. Need to provide a more detailed backgroud information on the current status of utilizing ADT + ARSI vs ADT or CAB. Also there need to be better definition of CAB. What does it entail?
Minor comments
1. Line 105-107 needs to be rewritten - authors just listed patient numbers as mortality
2. Line 170-179 should be deleted
3. Language editing by a native speaker should be considered.
Comments on the Quality of English LanguageSee above. Language editing by a native speaker should be considered.
Author Response
Dear Samuel C. Mok and reviewers,
Thank you for reviewing my manuscript.
I have revised my manuscript according to the reviewers’ suggestions, and I believe that your comments will be helpful for improving quality of my manuscript.
The revised parts are shown in red.
Please re-review my manuscript.
Thanks again, and then I appreciate so much your kind efforts.
Yours sincerely,
Hiroaki Iwamoto, MD, PhD
Iwamoto-h@med.kanazawa-u.ac.jp
Department of Integrative Cancer Therapy and Urology
Kanazawa University Graduate School of Medical Science
Takaramachi, Kanazawa, Ishikawa, Japan
TEL: 076-265-2393, FAX 076-222-6726
Reviewer's comments:
<Reviewer: 3>
This is an interesting paper that aimed to propose a cost-effective treatment strategy that also potentially avoids overtreatment. Based on outcome data from published studies and local patients, the authors applied the Canazawa criteria and stratified patients into low and high risk groups, and proposed in the low risk group a subset of patients can be safely kept on ADT before adding ARSI. Several issues need to be considered
Major comments
The authors stated patient can be treated with ADT and be switched to ARSI based on the rate of PSA reduction at 12 weeks. The only data was shown was patient had < 95% PSA reduction at 12 weeks has worse outcome. There is no data support by adding ARSI to these patients can improve their outcome comparing to only ADT alone.
(Reply)
Thank you for kindly comments.
Switching to ARSI at 12 weeks or less can be treated as upfront ARSI according to guideline recommendations.
The purpose of this study is to identify patients who do not need to be treated with upfront ARSI.
Basically, it is assumed that upfront ARSI (ARSI administered within 12 weeks) will be performed.
Performing upfront ARSI on all patients has the potential to overtreat, and there is a health economic benefit to selecting patients who do not need upfront ARSI.
- The authors need to clarify how the threshold of 95% reduction of PSA was determined.
(Reply)
Thank you for kindly comments.
The threshold at which survival was most clearly divided was 95% PSA reduction after 12 weeks.
- Figure 1 - total patients included in the study was 218. The sum of low risk and high risk patient numbers are less than 218 for the two trials and Canazawa criteria. Where are the missing patients?
(Reply)
Thank you for kindly comments.
The number of patients is small because patients for whom data for all three factors of the Canazawa classification were not available were excluded.
I followed your advice and added the following sentence. "Patients with missing data were excluded for risk classification." ( P4, lines 104-105)
- The introduction part needs to be expanded. Need to provide a more detailed backgroud information on the current status of utilizing ADT + ARSI vs ADT or CAB. Also there need to be better definition of CAB. What does it entail?
(Reply)
Thank you for kindly comments.
I have followed your advice and added and revised the introduction. ( P3, lines 53-61)
Minor comments
- Line 105-107 needs to be rewritten - authors just listed patient numbers as mortality
(Reply)
Thank you for kindly comments.
I followed your advice and revised the text
- Line 170-179 should be deleted
(Reply)
Thank you for kindly comments.
I followed your advice and deleted the text.
- Language editing by a native speaker should be considered.
(Reply)
Thank you for kindly comments.
I have followed your advice and have had the text edited by a native English speaker.
Comments on the Quality of English Language
See above. Language editing by a native speaker should be considered.
(Reply)
Thank you for kindly comments.
I have followed your advice and have had the text edited by a native English speaker.
Round 2
Reviewer 1 Report
Comments and Suggestions for Authors
Thanks
Author Response
Dear Samuel C. Mok and reviewers,
Thank you for reviewing my manuscript.
Your advice helped me to improve my manuscript.
Thanks again, and then I appreciate so much your kind efforts.
Yours sincerely,
Hiroaki Iwamoto, MD, PhD
Iwamoto-h@med.kanazawa-u.ac.jp
Department of Integrative Cancer Therapy and Urology
Kanazawa University Graduate School of Medical Science
Takaramachi, Kanazawa, Ishikawa, Japan
TEL: 076-265-2393, FAX 076-222-6726
Reviewer 2 Report
Comments and Suggestions for Authors
Dear authors,
Thank you for your kindly response to my concern. Unfortunately, the revised manuscript is still unreadable based on the following reasons:
1. Unclear unmet medical need. The authors explain that mCRPC patients potentially have risk in over-treatment due to ARSI+ADT in the response but not in the manuscript and which will cause the readers unable to know the novelty of this study. Additionally, the evolution of CAB from ADT+LHRH-A to ARSI+ADT is not mandatory to this study, which may interfere the description of the rationale. By the way, the tumor-controlling efficacy of ADT+LHRH-A had been discussed in a systemic review (please refer to PMID 30853808).
2. At Line 63-65, the authors disclosed "However, a certain number of cases have a long-term response to ADT or CAB (combined androgen blockade) in Asians, including Japanese patients." This sentence is confusing because which can not logically link to the following sentence "It is important from a medical and economic point of view to identify patients who have a significant response to ADT within 12 weeks after beginning ADT, which was the inclusion criterion for an ARSI clinical trial for mCSPC." Also, long-term of tumor-controlling efficacy of ADT or CAB to mCRPC patients is a good thing. Why do the authors use "However" here?
3. The Materials and Methods section should have segmentation. A pull description leads this section totally unreadable.
4. LATITUDE and CHAARTED refer to two randomized clinical trials regarding to ADT plus anti-inflammatory reagents (Please refer to PMID 28578607 and 29384722). What is "the LATITUDE and CHAARTED classifications" at Line 82 indicated? If the study data was from these two clinical trials, the authors should disclose it. If not, the authors should describe and cite the classification properly.
5. On the first paragraph of Discussion, the authors disclose the comparising efficacy among "Canazawa classification," "LATITUDE classification," and " CHAARTED classification." However, at the last few sentences of introduction, the authors disclosed that "It is important from a medical and economic point of view to identify patients who have a significant response to ADT within 12 weeks after beginning ADT, which was the inclusion criterion for an ARSI clinical trial for mCSPC." These two description can not be matched, indicated that the description in Result section is digressed.
Author Response
Dear Samuel C. Mok and reviewers,
Thank you for reviewing my manuscript.
I have revised my manuscript according to the reviewers’ suggestions, and I believe that your comments will be helpful for improving quality of my manuscript.
The revised parts are shown in red.
Please re-review my manuscript.
I have created manuscript, but if you have any problems please let me know.
Thanks again, and then I appreciate so much your kind efforts.
Yours sincerely,
Hiroaki Iwamoto, MD, PhD
Iwamoto-h@med.kanazawa-u.ac.jp
Department of Integrative Cancer Therapy and Urology
Kanazawa University Graduate School of Medical Science
Takaramachi, Kanazawa, Ishikawa, Japan
TEL: 076-265-2393, FAX 076-222-6726
Reviewer's comments:
<Reviewer: 2>
Dear authors,
Thank you for your kindly response to my concern. Unfortunately, the revised manuscript is still unreadable based on the following reasons:
- Unclear unmet medical need. The authors explain that mCRPC patients potentially have risk in over-treatment due to ARSI+ADT in the response but not in the manuscript and which will cause the readers unable to know the novelty of this study. Additionally, the evolution of CAB from ADT+LHRH-A to ARSI+ADT is not mandatory to this study, which may interfere the description of the rationale. By the way, the tumor-controlling efficacy of ADT+LHRH-A had been discussed in a systemic review (please refer to PMID 30853808).
(Reply)
Thank you for kindly comments.
The possible risk of overtreatment with ARSI+ADT is described in detail in the discussion. (P11-12, lines 251-287)
I have added that to the Introduction following your advice.
Regarding CAB, I have included it in the Introduction following reviewer3's advice.
It is possible to delete it, but I will abide by your and the editor's discretion.
- At Line 63-65, the authors disclosed "However, a certain number of cases have a long-term response to ADT or CAB (combined androgen blockade) in Asians, including Japanese patients." This sentence is confusing because which can not logically link to the following sentence "It is important from a medical and economic point of view to identify patients who have a significant response to ADT within 12 weeks after beginning ADT, which was the inclusion criterion for an ARSI clinical trial for mCSPC." Also, long-term of tumor-controlling efficacy of ADT or CAB to mCRPC patients is a good thing. Why do the authors use "However" here?
(Reply)
Thank you for kindly comments.
In Asians, including Japanese, there are a certain number of cases in which ADT or CAB provides a long-term response.
However, the guidelines recommend upfront ARSI, and not initiating treatment with ADT or CAB.
I used "however" because it is contradictory that upfront ARSI is recommended for all mCSPC, but there are patients who provide long-term responses to ADT or CAB.
As you pointed out, this was difficult to understand, so I have revised it.
I have also added sentences to make the continuity of the sentences easier to understand. (P3, lines 63-66)
- The Materials and Methods section should have segmentation. A pull description leads this section totally unreadable.
(Reply)
Thank you for kindly comments.
I followed your advice and split the materials and methods section into understandable sections. (P3-4, lines 71, 75, 93, 101)
I think your advice made it very easy to understand.
- LATITUDE and CHAARTED refer to two randomized clinical trials regarding to ADT plus anti-inflammatory reagents (Please refer to PMID 28578607 and 29384722). What is "the LATITUDE and CHAARTED classifications" at Line 82 indicated? If the study data was from these two clinical trials, the authors should disclose it. If not, the authors should describe and cite the classification properly.
(Reply)
Thank you for kindly comments.
The expression LATITUDE and CHAARTED classifications is incorrect, so we have revised it to LATITUDE and CHAARTED criteria. (P3, lines 86) (P4, lines 122, 124, 133-134, 136) (P6, lines 185, 186) (P7, lines 190, 191) (P11, lines 236, 243)
We have also added references for clarity. (P3, lines 86)
- On the first paragraph of Discussion, the authors disclose the comparising efficacy among "Canazawa classification," "LATITUDE classification," and " CHAARTED classification." However, at the last few sentences of introduction, the authors disclosed that "It is important from a medical and economic point of view to identify patients who have a significant response to ADT within 12 weeks after beginning ADT, which was the inclusion criterion for an ARSI clinical trial for mCSPC." These two description can not be matched, indicated that the description in Result section is digressed.
(Reply)
Thank you for kindly comments.
In this study, we examined all three risk classifications: Canazawa, LATITUDE, and CHAARTED.
In all low-risk, low-volume risk classifications, we showed that TTCR was significantly shorter when the PSA decline rate 12 weeks after initiating ADT was less than 95%.
Unlike LATITUDE and CHAARTED, the Canazawa risk classification is unfamiliar to readers.
I felt that treating the three risk classifications on the same level without showing the usefulness of the Canazawa risk classification would not gain readers' understanding.
Therefore, we felt it was necessary to show the usefulness of the Canazawa risk classification in comparison with LATITUDE and CHAARTED, and have described this in the Discussion.
Reviewer 3 Report
Comments and Suggestions for Authors
Thank you for addressing my previous comments.
In the response to my first comment, the authors mentioned "Switching to ARSI at 12 weeks or less can be treated as upfront ARSI according to guideline recommendations." Please provide reference to the guideline that the authors referred.
Author Response
Dear Samuel C. Mok and reviewers,
Thank you for reviewing my manuscript.
I have revised my manuscript according to the reviewers’ suggestions, and I believe that your comments will be helpful for improving quality of my manuscript.
Please re-review my manuscript.
Thanks again, and then I appreciate so much your kind efforts.
Yours sincerely,
Hiroaki Iwamoto, MD, PhD
Iwamoto-h@med.kanazawa-u.ac.jp
Department of Integrative Cancer Therapy and Urology
Kanazawa University Graduate School of Medical Science
Takaramachi, Kanazawa, Ishikawa, Japan
TEL: 076-265-2393, FAX 076-222-6726
Reviewer's comments:
<Reviewer: 3>
Thank you for addressing my previous comments.
In the response to my first comment, the authors mentioned "Switching to ARSI at 12 weeks or less can be treated as upfront ARSI according to guideline recommendations." Please provide reference to the guideline that the authors referred.
(Reply)
Thank you for kindly comments.
I apologize for writing a sentence that is difficult to understand.
I was referring to the inclusion criteria for ARSI clinical trials for mCSPC when I said, “Switching to ARSI at 12 weeks or less can be treated as upfront ARSI.”
The following guidelines recommend upfront ARSI.
References 6
Schaeffer, E.M.; Srinivas, S.; Adra, N.; An, Y.; Barocas, D.; Bitting, R.; Bryce, A.; Chapin, B.; Cheng, H.H.; D'Amico, A.V.; et al. Prostate Cancer, Version 4.2023, NCCN Clinical Practice Guidelines in Oncology. J Natl Compr Canc Netw 2023, 21, 1067-1096, doi:10.6004/jnccn.2023.0050.